# Solar flare accelerates nearly all electrons in a large coronal volume

Gregory D. Fleishman[1✉], Gelu M. Nita[1], Bin Chen[1], Sijie Yu[1] & Dale E. Gary[1]

Solar flares, driven by prompt release of free magnetic energy in the solar corona[1,2], are known to accelerate a substantial portion (ten per cent or more)[3,4] of available electrons to high energies. Hard X-rays, produced by high-energy electrons accelerated in the flare[5], require a high ambient density for their detection. This restricts the observed volume to denser regions that do not necessarily sample the entire volume of accelerated electrons[6]. Here we report evolving spatially resolved distributions of thermal and non-thermal electrons in a solar flare derived from microwave observations that show the true extent of the acceleration region. These distributions show a volume filled with only (or almost only) non-thermal electrons while being depleted of the thermal plasma, implying that all electrons have experienced a prominent acceleration there. This volume is isolated from a surrounding, more typical flare plasma of mainly thermal particles with a smaller proportion of non-thermal electrons. This highly efficient acceleration happens in the same volume in which the free magnetic energy is being released[2].

The microwave analysis is performed using imaging spectroscopy data from the Expanded Owens Valley Solar Array (EOVSA) described in detail elsewhere[2]. We use independent pixel-to-pixel and time-to-time spectral model fitting of these microwave imaging spectroscopy data to obtain evolving, spatially resolved distributions (maps) of suprathermal and thermal electrons. These maps pinpoint the location and shape of the evolving acceleration region in a large solar flare that occurred on 10 September 2017 (Fig. 1). This now famous flare has attracted extraordinary attention as it demonstrates several 'textbook' flare properties, which were observed with unprecedented coverage and resolution across the electromagnetic spectrum[7–14].

Figure 2 shows an example of these parameter maps for one time frame; the entire evolution is illustrated in Supplementary Video S1. Figure 2 also shows two regions of interest (ROIs), ROI1 and ROI2, kept fixed for all analysed time frames, which inscribe two areas having the most reliable spatially resolved spectra and, thus, the most reliable model spectral fitting diagnostics (see Methods). ROI1 inscribes the area in which the fast and strong release of coronal magnetic energy has been measured[2], whereas ROI2 is a reference area of more typical flare plasma, outside the acceleration region, to be used for comparison.

We focus on ROI1, in which the fast, strong release of magnetic energy occurred during the main flare phase[2], thus pinpointing the exact energy release region. Figure 2d shows that ROI1 inscribes an extended area (corresponding to an estimated volume of about $1.67 \times 10^{27}$ cm$^3$; see Methods), in which the number density of suprathermal electrons with high energies above 20 keV is very large—up to around $10^{10}$ cm$^{-3}$. By contrast, the number density of thermal electrons in ROI1, shown in Fig. 2c, is undetectably small (see Methods): the map contains an extended thermal density 'hole' roughly coinciding with ROI1. This directly implies that the number density of suprathermal electrons is much larger than that of the thermal electrons in the region in which the release of magnetic energy takes place.

Supplementary Video S1 demonstrates that the gap in the thermal electron distribution holds for the entire duration of the analysed four-minute episode around the peak of the flare, although its shape evolves and shows an overall outward motion (to the right in the figure), and it continues to match the region of enhanced suprathermal electron density. These spatio-temporal evolutions show that, during the entire episode, ROI1 and ROI2 differ fundamentally in character: suprathermal electrons dominate in ROI1, whereas the thermal electrons dominate in ROI2. The suprathermal electrons in ROI1 seem to have been accelerated in place, rather than transported there from elsewhere (see Methods). Therefore, ROI1 combines three properties: (1) fast release of a large amount of magnetic energy[2]; (2) depletion of thermal plasma; and (3) presence of a dense population of suprathermal electrons, presumably accelerated owing to the magnetic energy release. This combination of properties implies that we have resolved the heart of the solar flare—the exact acceleration region that places strong constraints on the physical mechanism driving the acceleration of electrons in the flare. Indeed, any mechanism capable of producing a suprathermal particle population has to extract a fraction of charged particles from the thermal plasma pool and increase individual energies of those particles greatly. As a result, at this acceleration stage, the number of accelerated suprathermal particles increases at the expense of the thermal particles, whose number density proportionally decreases. In our case, ROI1 has a lower (possibly much lower) than 10% proportion of thermal plasma in a region of high suprathermal electron density (see Methods). This means that a large fraction, essentially all, of the thermal electrons originally present in this volume has been converted to the suprathermal electron population during (and, presumably, owing to) this energy release. We conclude that the magnetic energy release in the solar flare offers a highly efficient engine for particle acceleration, which is capable of converting essentially all ambient electrons with thermal energies (for example, less than ≈1 keV) into a

[1]Center for Solar-Terrestrial Research, New Jersey Institute of Technology, Newark, NJ, USA. ✉e-mail: gfleishm@njit.edu

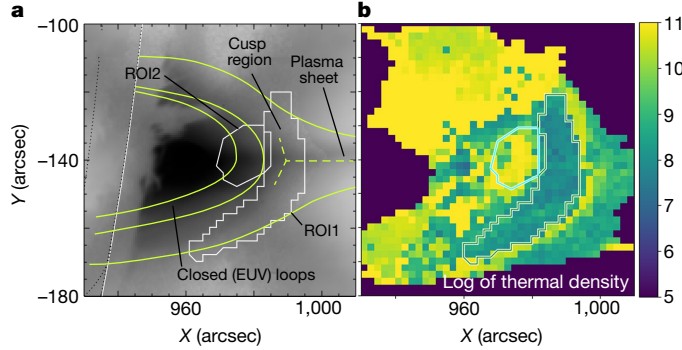

**Fig. 1 | Thermal plasma in the context of magnetic morphology in the 10 September 2017 solar flare. a**, The plot shows EUV brightness distribution (image) in the context of the hand-drawn magnetic field lines outlining closed, post-reconnection loops, cusp region including a so-called Y-point and the vertical plasma sheet. ROI1 and ROI2 used in the analysis are also shown. **b**, Distribution of the median values of the thermal plasma density, log scale, obtained from the microwave data using MCMC simulations; see Methods. Strong depletion of the thermal number density is apparent inside ROI1.

suprathermal population of electrons with high energies exceeding 20 keV (see Methods).

The release of free magnetic energy, quantified by the fast decay of the magnetic field at the rate $\dot{B} \approx 5\,\text{G}\,\text{s}^{-1}$, has been suggested to be driven by turbulent magnetic reconnection within an extended volume of the cusp region of the flare[2]. This is motivated by the inferred highly enhanced turbulent magnetic diffusivity, $\nu \approx 10^{15}\,\text{cm}^2\,\text{s}^{-1}$, and the associated strong electric field, $E \approx 20\,\text{V}\,\text{cm}^{-1}$, in that extended volume[2].

The fundamental force capable of producing work on charged particles is the electric force. The acceleration efficiency is specified by a balance between the energy gain due to the electric field and the energy loss through collisions, which defines a critical value of the electric field, called the Dreicer field, $E_D$ (ref. [15]). The condition for runaway acceleration is $E \gg E_D$, which is called a 'super-Dreicer' electric field. The electric field inferred from the magnetic field decay[2], about $20\,\text{V}\,\text{cm}^{-1}$, is many orders of magnitude larger than the estimated Dreicer field, which is $E_D \approx 10^{-4}\,\text{V}\,\text{cm}^{-1}$.

To support the simultaneous acceleration of literally all ambient electrons in a macroscopic volume such as ROI1, this strong super-Dreicer field must be present over a substantial portion of ROI1. As noted above, this is consistent with the observed simultaneous decay of the magnetic field over the entire ROI1 (ref. [2]), indicative of turbulent magnetic reconnection, in which the dissipation of the magnetic energy takes place throughout the volume. This is in contrast to the alternative view that all acceleration takes place in one or a few isolated points (X-points or O-points[16]) favourable for macroscopic reconnection.

From this work, there emerges a consistent picture of particle acceleration in the magnetic energy release region: (1) the decay of the magnetic field owing to turbulent magnetic reconnection produces a strong super-Dreicer electric field over an extended volume; (2) this strong electric field works over literally all ambient particles, which boosts their energies up to 20 keV and higher; (3) this acceleration process is so efficient and persistent that it does not leave any measurable thermal plasma component compared with the highly dominant suprathermal component.

Models of particle acceleration owing to magnetic reconnection, including 2D and 3D particle-in-cell simulations, as well as a new large-scale kinetic simulation approach kglobal[17,18], suggest that efficiency of the acceleration is linked to a ratio of the reconnecting

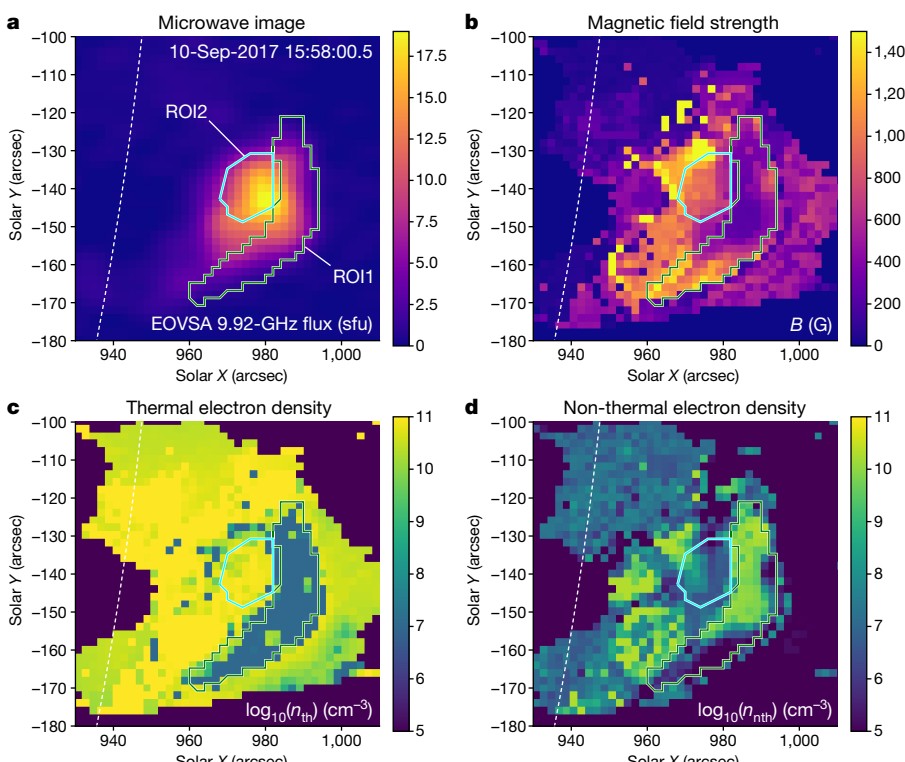

**Fig. 2 | Spatial distributions of flare parameters. a**, EOVSA map at 9.92 GHz taken at 15:58 UT. **b**–**d**, Maps of the magnetic field (**b**), thermal plasma density (**c**) and non-thermal plasma density (**d**) derived for the same time from the bulk model spectral fitting (see Methods). ROI1 inscribes the hole in the distribution of thermal plasma, which also corresponds to a peak in the number density of the suprathermal electrons. ROI2 inscribes a reference area; see Methods. The dotted arc shows the solar limb. Note that, owing to different processing, panel **c** differs slightly from Fig. 1b, which was produced by the more thorough but time-consuming MCMC method; see Methods. sfu, solar flux unit.

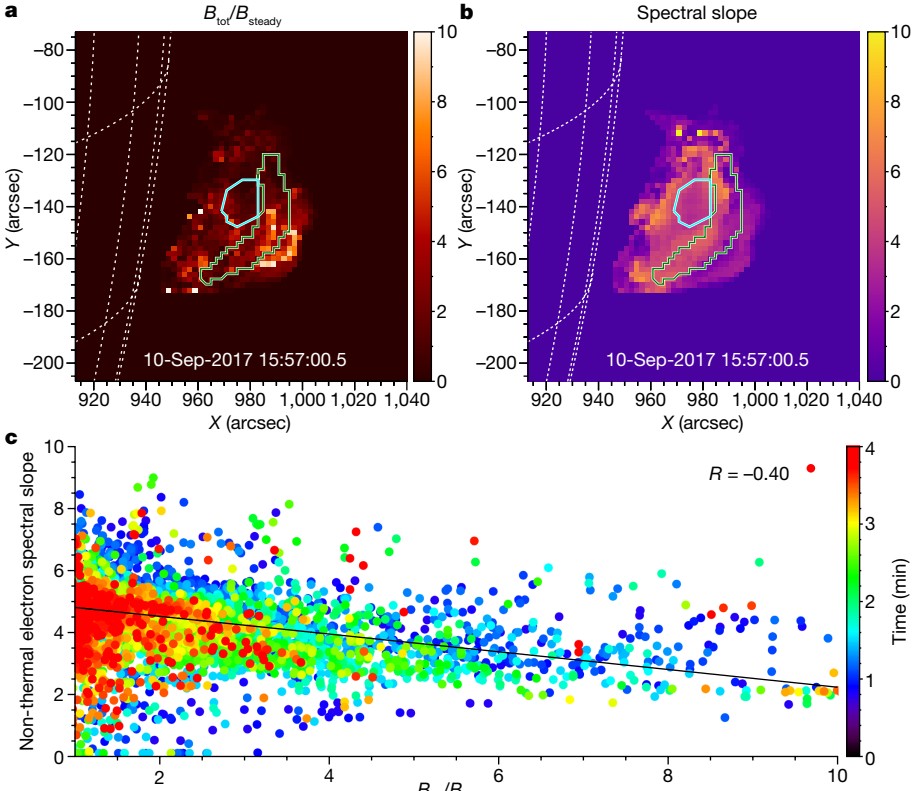

**Fig. 3 | Spectral slope of the suprathermal component versus reconnecting magnetic field. a**, **b**, Maps of the inferred $r_B = B_{tot}/B_{steady}$ ratio (**a**) and the suprathermal electron energy spectral slope (**b**), averaged over the first five time frames (20 s) of the full four-minute interval. The ROI1 and ROI2 regions are marked, respectively, by green and cyan contours in both panels. **c**, Dot symbols show the spectral index $\delta$ obtained from the ROI1 model fit for each pixel and time versus the running value of the $r_B$ ratio. The correlation data are plotted for the analysed four-minute interval, colour coded as time increases, as shown by the colour scale. The solid line shows the linear fit to the data over the first two minutes, when most of the energy release takes place, which corresponds to a correlation coefficient $R = -0.40$. An animated version of this figure (Supplementary Video S2) showing all time frames used in this analysis is provided as supplemental material.

(dissipating) component $B_{rec}$ of the total magnetic field and the remaining component, called the guide field, $B_g$ (ref. [19]), does not explicitly take part in the reconnection. According to the models[17,18,20–22], efficient acceleration requires that $B_{rec}$ is larger than $B_g$. We check this expectation with our data. Although we cannot properly separate these two components observationally, we can compare the observed total magnetic field, $B_{tot}(t)$, at a given time and location with its value $B_{steady}$ near the end of the decay period, when it becomes steady[23]. We estimate $B_{steady}$ at each location in ROI1 as the mean $B$ evaluated over the last 20 s of the four-minute episode[2]. $B_{steady}$ serves as an estimate for the magnetic field component that does not participate in the energy release process, which includes $B_g$. Then, for each pixel, we form a ratio $r_B(t) = B_{tot}(t)/B_{steady}$, in which $B_{tot}(t)$ is the instantaneous value of the magnetic field inferred from the spectral model fit, which includes both decaying and steady components of the magnetic field. If the observed $r_B \gg 1$, then it is probable that $B_{rec}/B_g \gg 1$ as well (small guide field case); thus, $r_B \gg 1$ could be viewed as a good proxy for efficient acceleration. We focus on the first two minutes of our four-minute episode, in which the condition $r_B \gg 1$ holds for many pixels and times.

We investigate the relationship between parameters $n_{nth}$ and $\delta$ of the suprathermal electron component derived from the spectral fit and the ratio $r_B$, in which the non-thermal number density exceeds the thermal number density. We did not find any correlation of the number density of the suprathermal electrons with $r_B$. This is consistent with the observed strong efficiency of the acceleration, which results in virtually all ambient electrons being accelerated. What is

correlated with acceleration efficiency is the power law spectral index $\delta$, as shown in Fig. 3. The correlation is such that a larger $r_B$ (proxy for the small guide field case) implies a smaller spectral index (harder energy spectrum), thus validating the theoretical expectations[18]. A simplistic interpretation of this relationship is that having more free magnetic energy (larger $r_B$) permits acceleration to higher energies, thus producing a flatter distribution of the accelerated electrons over energy.

In addition, our observations show that the suprathermal electrons, generated in a region in which virtually all ambient electrons are accelerated, remain almost perfectly isolated from the surrounding cooler plasma for a time period much longer than the source transit time, even though the system does not contain any 'solid walls' that would hold suprathermal particles in. This means that the system contains a highly efficient physical process or magnetic topology that traps the suprathermal particles within the volume they occupy. Otherwise, the suprathermal particles would become much more uniformly mixed with the ambient thermal particles, which is not observed. An important process capable of providing this trapping is enhanced angular diffusion that reduces the particle mean free path[24]. Such diffusion is due to particle scattering by the turbulent magnetic field, which is also responsible for acceleration of the particles. Although the need for this enhanced diffusion is strongly suggested by the observations, the important characteristics of the corresponding turbulent magnetic field, such as their spectral and spatial structure and evolution, remain unknown and call for dedicated modelling.

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

## Methods

### Overview

In this study, we used the dataset from the EOVSA[25] described in an earlier paper[2]. The model spectral fitting, its parameters and their uncertainties were described in the supplementary materials to that paper. The parameters used to create the evolving maps of the thermal and suprathermal electrons in the flare region are from the same spectral fits as those used for the magnetic field maps reported there. Here we used these maps of electron parameters to investigate the spatially resolved structure and evolution of the electron acceleration in the spatial area that showed the most prominent decay of the coronal magnetic field[2].

### Spatially resolved microwave spectra and selection of ROIs

Extended Data Figure 1 shows a representative set of the observed spatially resolved microwave spectra from pixels with an area of 2″ × 2″ (about $2.1 \times 10^{16}$ cm$^2$) and associated model spectral fits distributed over the flare region. For reference, the central panel shows a single microwave image at 9.92 GHz taken at 15:58 UT, which corresponds to the main peak of the flare. (For the microwave image, the instrumental beam is 113″.7/$f$ (GHz) × 53″.0/$f$ (GHz). A circular restoring beam with full width at half maximum of 87″.9/$f$ (GHz) was used, which is about 9″ for 9.92 GHz shown in the figure.) The spectral fitting uses the model of the gyrosynchrotron source function with the account of the free–free component[2]. We performed this model spectral fitting over all 60 time frames and over all pixels in these 60 map cubes, assuming a source depth along the line of sight (LOS) of 5.8 Mm (this corresponds to 8″ on disk, which is a scale of features (loops) seen in the flare images). The primary ROI, ROI1, indicated by the green contour, includes 137 image pixels that, under the same LOS depth assumptions, correspond to an estimated volume of about $1.7 \times 10^{27}$ cm$^3$. Consequently, the reference ROI, ROI2, shown by the cyan contour, which encloses 49 pixels, corresponds to an estimated volume of around $6.0 \times 10^{26}$ cm$^3$. The numbered points are pixels whose spectra and fits are shown in the other eight panels of the plot.

ROI1 inscribes the area in which the most prominent decay of the magnetic field has been detected, a small portion of which was analysed in the earlier paper[2]. Here we analyse the entire ROI1 as it shows a coherent depletion of the thermal plasma and a high density of suprathermal electrons. The spatially resolved spectra (for example, pixels P1 and P4) from an upper portion of ROI1 have high signal-to-noise ratio and their spectral peaks occur within the frequency range observed by the EOVSA. As a result, the model spectral fitting diagnostics using such spectra are the most robust (see the next section). In the bottom portion of ROI1, the spectra have lower signal-to-noise ratio (see example in pixel P6), especially at high frequencies, which can result in larger uncertainties of the spectral index that quantifies the suprathermal electron distribution over energy (see supplemental materials in the earlier paper[2]).

In the reference area ROI2, the signal-to-noise ratio is also high. The spectral peak is outside the EOVSA frequency range, indicative of high magnetic field in ROI2. The model spectral fitting of such spectra typically yields a reliable estimate of the thermal number density, whereas the magnetic field and suprathermal electrons are recovered with larger uncertainties (see the next section).

Four other spectra from the figure corners show spectra from pixels P3, P5, P7 and P8. The signal-to-noise ratios are not large there; however, the fits are within the uncertainties and the spectra show expected trends: the spectral peak frequency is high from P3 and P8 locations close to the solar limb (which means high magnetic field strength), whereas the peak frequency is lower from higher locations P5 and P7 (which implies lower magnetic field strength). We note that because of high uncertainties of the data in the four 'corner' cases, the uncertainties of the derived physical parameters are also large there. Although we

present parameters from all these fits in Fig. 2, we restrict our quantitative analysis to the most reliable spectra and fits from ROI1 and ROI2 and, hence, those four spectra are excluded.

### MCMC validation of the spectral model fit

The main reported result, that the number density of high-energy electrons is much larger within ROI1 than that of the thermal plasma, is based on the model spectral fitting of the microwave data. Here we use the Markov chain Monte Carlo (MCMC) simulations, implemented by an open-source Python package emcee[26], to derive statistical distributions of the model fit parameters to quantify the confidence of this finding. This approach explores the full multidimensional space of the model fit parameters to both provide parameter distributions and show correlations between them. For this reason, it is much more time consuming than the speed-optimized GSFIT approach[2], with which the bulk model spectral fitting has been performed. We restrict our MCMC analysis to all pixels in a single time frame, the same as shown in Fig. 2, which takes considerably longer than the GSFIT analysis of the entire 60-frame time sequence, but a comparison of the MCMC result in Fig. 1b with the bulk fitting in Fig. 2c shows that the results are comparable and fully consistent.

Thermal and suprathermal electrons affect the microwave spectrum differently. The suprathermal electrons gyrating in the ambient magnetic field are responsible for generation of the microwave emission. In the optically thin regime (high frequencies), the contributions of each individual electron add up incoherently; thus, the microwave flux level of the emission is proportional to the number density of the suprathermal component. In the optically thick regime (low frequencies), the flux of the microwave emission is determined by the energy of the electron population responsible for the emission at a given frequency. For these reasons, the microwave diagnostics of the suprathermal electrons is robust, provided that both low-frequency and high-frequency spectral ranges are available.

The thermal electrons contribute much less to the radiation intensity. Their main effect on the microwave radiation spectrum is due to dispersion of electromagnetic waves; simplistically speaking, due to the index of refraction. In the plasma, the index of refraction depends on the plasma frequency $\omega_p$, which is defined by the number density of the ambient free electrons:

$$\omega_p^2 = \frac{4\pi e^2 n_{tot}}{m},$$  (1)

in which $e$ and $m$ are the charge and mass of the electrons and $n_{tot}$ is the total number density of all ambient free electrons—both thermal $n_{th}$ and suprathermal $n_{nth}$:

$$n_{tot} = n_{th} + n_{nth}.$$  (2)

As $n_{th}$, and thus $n_{tot}$, increases, the microwave flux decreases at low frequencies, as illustrated in Supplementary Video S3. Thus, the diagnostic of $n_{th}$ is primarily based on the microwave spectral shape at low frequencies. If $n_{th} \gg n_{nth}$, then $n_{tot} \approx n_{th}$, offering the diagnostics of the thermal electron number density.

The MCMC analysis of a spectrum (from a pixel inside ROI2) that yields a well-constrained thermal number density is shown in Extended Data Fig. 2. The figure layout is as follows. The stand-alone upper-right panel shows a measured spectrum from a pixel within ROI2 (open circles with error bars) and a set of theoretical trial spectra (blue) consistent with the data. The panels placed over the diagonal show statistical distributions (histograms) of the trial fits for all six model parameters. The remaining panels show correlations between all possible pairs of these parameters. In this case, the distribution of thermal plasma number density is very narrow; thus, this parameter is well constrained (see also the next section). This is due to the well-measured low-frequency

part of the spectrum, whose deviation from a simple power law permits this thermal density diagnostic, as explained above. By contrast, other parameters have broader statistical distributions and, thus, they are not that well constrained. This is due to the absence of the optically thin part of the measured spectrum, because the spectral peak extends beyond the EOVSA frequency range. Although the distribution of the suprathermal electron number density is broad, its relatively low most-probable value is consistent with the dominance of the thermal electrons, $n_{th} > n_{nth}$.

The case when $n_{th} \ll n_{nth}$ is more problematic for the thermal plasma diagnostics, because now $n_{tot} \approx n_{nth}$ and the thermal plasma density is defined by the difference:

$$n_{th} = n_{tot} - n_{nth} \ll n_{tot}, \qquad (3)$$

which is the intrinsically less constrained given uncertainties of the inputs. Thus, if the contribution of the suprathermal electrons to the total ambient density dominates, it is problematic to obtain well-constrained values of the thermal number density separately. In such a situation, we can only confidently conclude that $n_{th} \ll n_{nth}$, which would—in fact—confirm that most of the available ambient electrons have been accelerated to high energies. The results of the MCMC simulations for a pixel from ROI1 are shown in Extended Data Fig. 3, which has the same layout as Extended Data Fig. 2. Here the spectrum contains the peak. The distributions of the magnetic field, suprathermal electron density and their spectral index are narrow; thus, these parameters are well constrained. The suprathermal electron number density is high, on the order of $n_{nth} \approx 10^{10}$ cm$^{-3}$. By contrast, the distribution of the thermal plasma number density is broad. It favours low $n_{th}$ values, falling steeply for higher values. These distributions show that the thermal density contribution to the total ambient number density $n_{tot}$ is undetectable compared with the non-thermal one, thus confirming that $n_{th} \ll n_{nth}$: the median values of $n_{th}$ are less than 5–10% of $n_{nth}$ and even the upper limit values computed as $n_{th} + 1\sigma_{n_{th}}$ are less than about 30% of $n_{nth}$ at many pixels within ROI1.

The maps of the thermal and suprathermal electron densities obtained from the MCMC simulations for the entire field of view are shown in Extended Data Fig. 4. They agree within the uncertainties with those obtained using GSFIT in Fig. 2. This confirms the reliability of the results derived using the fast model spectral fitting method used in GSFIT. One apparent disagreement between Extended Data Fig. 4a and Fig. 2c is the thin line of enhanced thermal density just to the right from ROI1 in the MCMC case. Although this feature is also present in Fig. 2c, it is made less apparent because the density falls less steeply, extending the light yellow colours higher in altitude and reducing the contrast. The reason for this different appearance of the maps is that Fig. 2c shows the most probable parameter value from the GSFIT analysis, whereas Extended Data Fig. 4a shows the median value from the corresponding statistical distribution of the parameter from the MCMC simulations (compare Extended Data Fig. 3 and Extended Data Fig. 2). When the uncertainties of the derived parameters are small (their statistical distribution is narrow), then the GSFIT value is very close to the median MCMC value. However, in the area to the right of ROI1, uncertainties of the derived parameters are larger, resulting in the different appearance of these maps, even though the values are consistent with each other within uncertainties, as has been said.

Extended Data Figure 4c illustrates the dominance of the suprathermal component in ROI1 by showing $\log(n_{th,max}/n_{nth})$, in which $n_{th,max}$ is represented as the median value of $n_{th} + 1\sigma_{n_{th}}$ of $n_{th}$ in MCMC. A diverging colour map is selected for this plot, in which white colour means $\log(n_{th,max}/n_{nth}) = 0$. The blue/white region shows up as a distinctive feature of ROI1, with the ratio $\log(n_{th,max}/n_{nth})$ ranging from 10% to 30%.

Note that the non-thermal number density $n_{nth}$ is sensitive to the value of the low-energy cut-off $E_{min}$, which we adopted to be fixed at 20 keV in GSFIT. In our MCMC test, we allow this parameter to vary. The assumption that $E_{min} = 20$ keV is proved valid in most regions of the map except in ROI1 (see the map of MCMC constrained $E_{min}$ in Extended Data Fig. 4d), in which the median values of $E_{min}$ reach 40–50 keV (see the sensitivity of the gyrosynchrotron spectrum to $E_{min}$ in Supplementary Video S3). Although such a concentration of non-thermal electrons can be owing to either acceleration in place or confinement of a transported electron population from elsewhere (for example, the X-point above)[11], the map of $E_{min}$ shows that it is about two times larger in ROI1 than in the surroundings, which is rather difficult to account for without bulk electron acceleration in ROI1. The simultaneous decay of magnetic field in this same region is further support for this. We thus conclude that the suprathermal electrons in ROI1 not only have a higher number density $n_{nth}$ but are also accelerated in bulk to a higher energy well separated from the thermal, Maxwellian component. In general, having larger $E_{min}$ may imply smaller $n_{nth}$ for the same spectral slope. However, the cross-correlation plots between the parameters shown in the bottom row of Extended Data Fig. 2 demonstrate that $E_{min}$ correlates with $\delta$ in such a way that larger $E_{min}$ corresponds to larger $\delta$ (softer spectra). As a result of this correlation, $n_{nth}$ does not correlate with $E_{min}$; thus, the conclusion of the high non-thermal number density is robust and does not depend strongly on the particular choice of $E_{min}$.

## A consistency check: comparison of microwave and EUV diagnostics of the coronal thermal plasma

A well-established way of investigating thermal coronal plasma is using extreme ultraviolet (EUV) emission, which is a combination of line emission from ions, primarily iron, in various ionization states (and, thus, is temperature-sensitive) and a continuum owing to bremsstrahlung. Here we use EUV data taken by the Solar Dynamics Observatory Atmospheric Imaging Assembly (SDO/AIA) in six narrow passbands sensitive to EUV emission from the corona. For each pixel within the field of view that we used to analyse the microwave emission, we applied a regularized differential emission measure (DEM) inversion[27] technique, from which we derived the emission measure (EM = $\int_{LOS} n_{th}^2 dL$, in which d$L$ is the differential column depth along the LOS) as a moment of the DEM. The thermal number density is then estimated as $n_{th} = \sqrt{EM/L}$, in which $L$ is 5.8 Mm, as adopted for the microwave spectral model fitting. The EM distribution is shown in Extended Data Fig. 5a. Owing to rather strong EUV emission, the EM map contains saturated areas and diffraction artefacts. Therefore, for quantitative analysis, we selected a small rectangular area within ROI2 that avoids these artefacts to the extent possible.

Direct pixel-to-pixel comparison, even in the case of a perfect co-alignment, would be inconclusive in our case for the following reasons: (1) the pixel sizes of the AIA and EOVSA maps are different (0.6″ and 2″, respectively); (2) the time cadence of data used for the analysis are different (12 s and 4 s, respectively). Therefore, we compare statistical distributions, rather than individual values, of the thermal electron number density obtained from these two different datasets.

We consider a single 12-s time range of the AIA data in a small rectangle area, marked in dark blue in Extended Data Fig. 5, free from strong artefacts, which contains 100 AIA pixels, and three 4-s time ranges of the EOVSA data in ROI2 that contains 49 pixels, giving a total of 147 measurements over the same 12-s time range. The standard DEM inversion techniques assume the so-called coronal elemental abundances, for which the Fe abundance is four times larger than in the photosphere. It was reported[28,29], however, that—in flaring volumes—the abundance can be closer to the photospheric one, owing to the fact that the thermal plasma is mainly due to chromospheric evaporation of material with photospheric abundance initiated by the precipitation of flare-accelerated particles into the chromospheric footpoint. Therefore, we used the AIA thermal plasma diagnostics assuming alternately both the coronal and the photospheric abundance. Another possible source of uncertainty of the EUV diagnostics is an assumption of ionization equilibrium, which can be strongly violated during

non-equilibrium flaring conditions. In addition, the EUV diagnostics suffer more from potential contributions along a long LOS (owing to the dependence of the EM on the column depth) compared with the microwave diagnostics, which are restricted to the region inside the non-thermal gyrosynchrotron source only.

With all these reservations in mind, Extended Data Fig. 5b shows a histogram of the thermal number density from the described rectangular ROI assuming the coronal abundance in filled dark blue and the photospheric abundance in empty dark blue. The filled light blue histogram shows the distribution of the thermal electron number density obtained for the three time frames for the entirety of ROI2. These distributions agree with each other within a factor of two (less for the photospheric abundance case), confirming that the thermal electron number densities derived from the microwave diagnostics in ROI2, in which they are statistically well constrained, are consistent with the EUV-derived numbers. We cannot perform a similar exercise in ROI1 because the microwave diagnostics of $n_{th}$ does not offer well-constrained values.

## Data availability

All original EOVSA data are maintained on the EOVSA website at http://www.ovsa.njit.edu/. Original EOVSA data used for this study are available at http://www.ovsa.njit.edu/fits/IDB/20170910/IDB20170910155625/. Fully processed EOVSA spectral imaging data in IDL save format can be downloaded from http://ovsa.njit.edu/publications/fleishman_ea_science_2019/data/.

## Code availability

All the codes we used in this study are based on publicly available software packages: GSFIT is available in the community-contributed SolarSoftWare repository, under the packages category, at www.lmsal.com/solarsoft/ssw/packages/gsfit/; the open-source MCMC code is documented in ref. [26].

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

**Acknowledgements** We thank the scientists and engineers who helped design and build the EOVSA, especially G. Hurford, S. White, J. McTiernan, W. Grammer and K. Nelin. This work was supported in part by NSF grants AGS-2121632, AGS-1817277, AST-1910354, AST-2130832, AGS-1654382 and AST-2108853 and NASA grants 80NSSC18K0667, 80NSSC20K0627, 80NSSC20K1318, 80NSSC21K0623, 80NSSC19K0068 and 80NSSC18K1128 to New Jersey Institute of Technology.

**Author contributions** G.D.F. developed the model spectral fitting methodology, participated in model fitting and analysis of the results and wrote the draft manuscript. G.M.N. developed the GSFIT spectral fitting package and participated in model fitting and analysis of results. B.C. developed the microwave spectral imaging and self-calibration strategy and adopted the MCMC methodology to perform the testing of the model fitting validity. S.Y. implemented the code of the microwave imaging pipeline under the guidance of D.E.G. and B.C. and performed the consistency check between the microwave and EUV diagnostics. D.E.G. led the construction and commissioning of the EOVSA and developed the observational strategy and calibration for microwave spectroscopy. All authors discussed the interpretation of the data, contributed scientific results and helped prepare the paper.

**Competing interests** The authors declare no competing interests.

**Additional information**
**Correspondence and requests for materials** should be addressed to Gregory D. Fleishman.

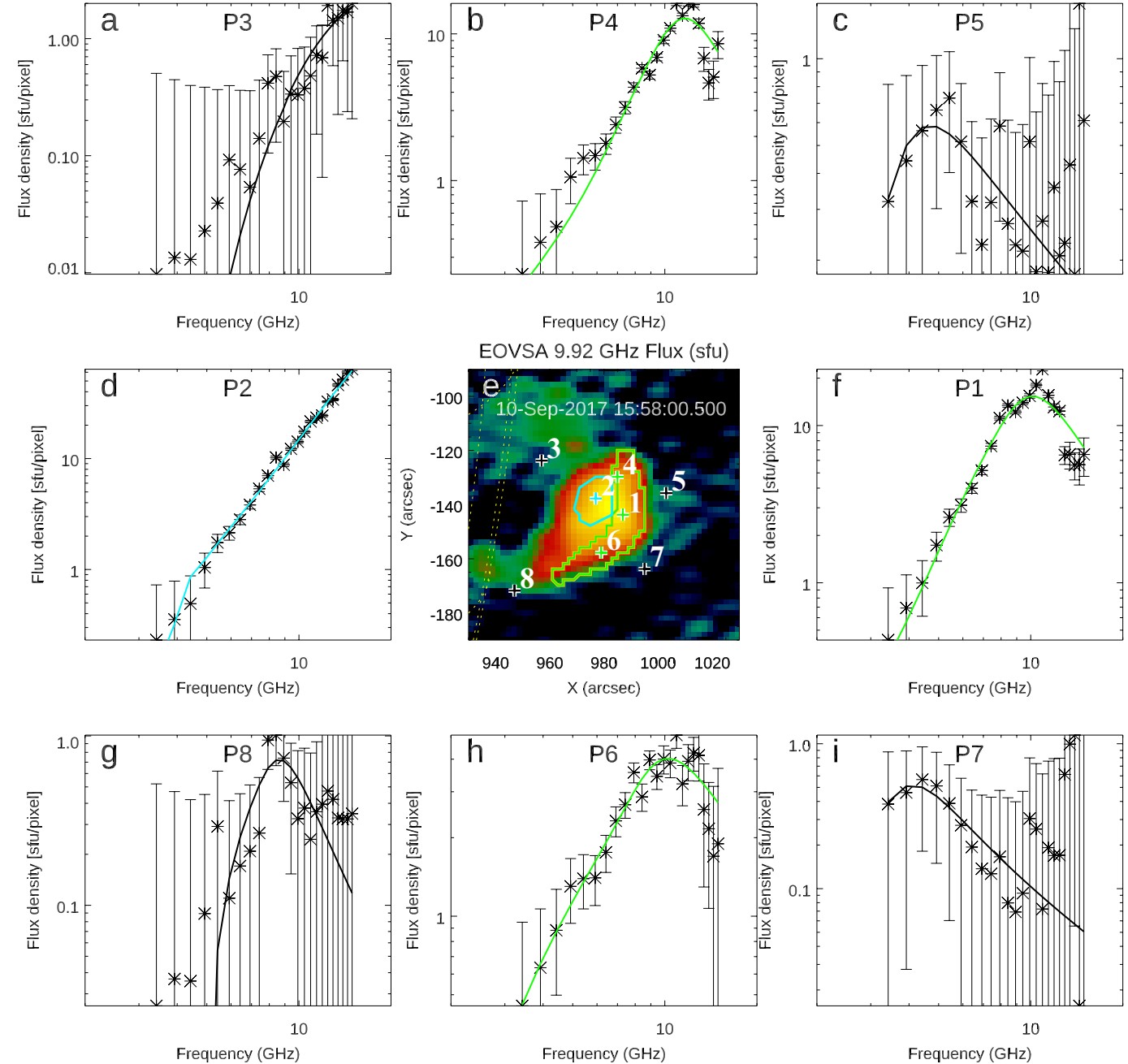

**Extended Data Fig. 1 | Example spectra from various locations and associated model fits.** The central panel (**e**) shows a reference microwave image at 9.92 GHz at 15:58 UT. ROI1 and ROI2 are shown in green and cyan, respectively. Plus signs, 1 to 8, indicate pixels, for which the observed spectra and model fits are shown in the remaining eight panels (**a**–**d**, **f**–**i**), each marked with P1 to P8. The asterisks with error bars show the data and the uncertainties at the 1-sigma level; the curves show the corresponding model spectral fits. The fluxes are given in solar flux units (sfu): 1 sfu = $10^4$ Jy = $10^{-22}$ W m$^{-2}$ Hz$^{-1}$ = $10^{-19}$ erg s$^{-1}$ cm$^{-2}$ Hz$^{-1}$.

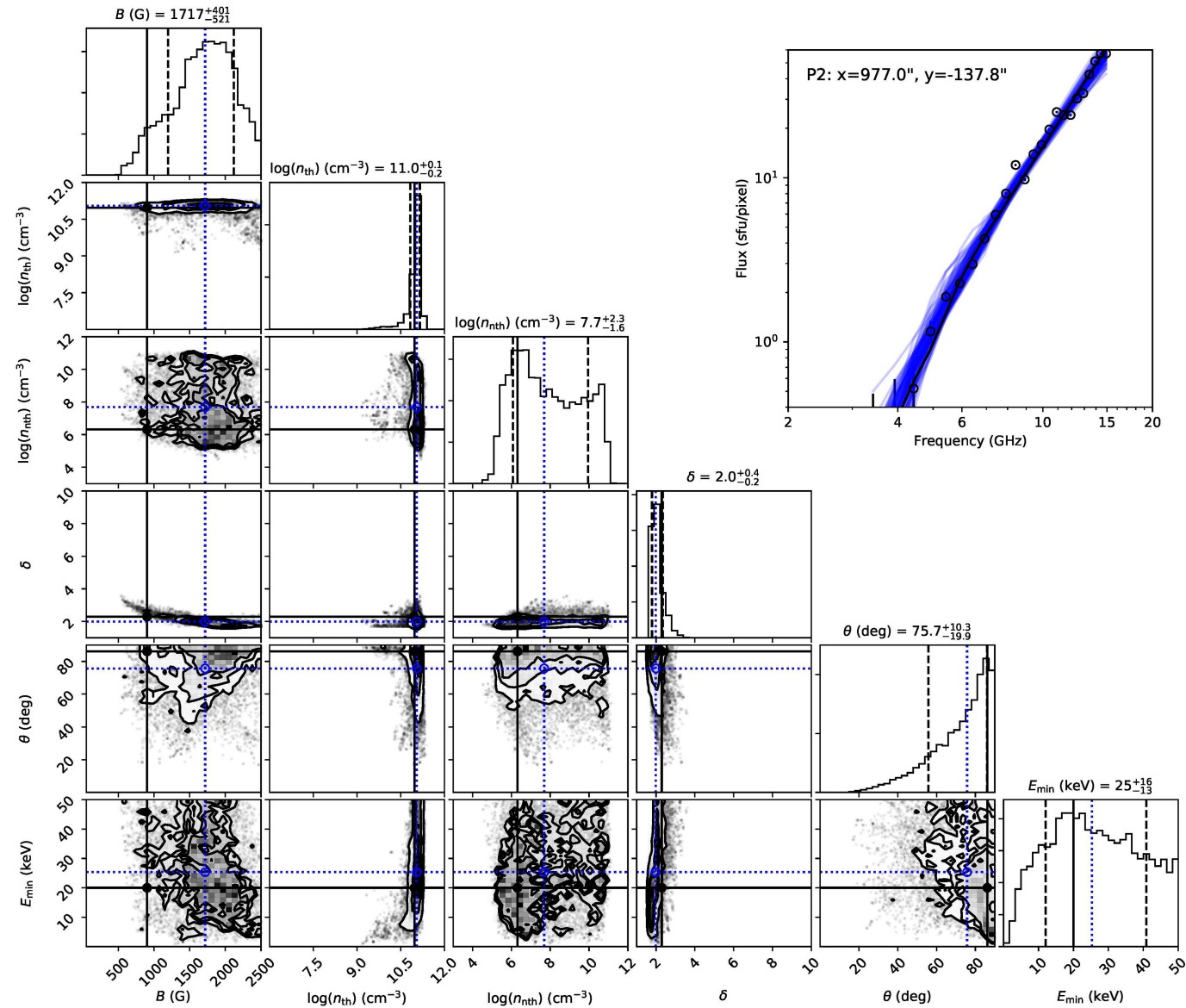

**Extended Data Fig. 2 | MCMC probability distributions of the fit parameters for an example pixel in ROI2.** The results are for pixel P2 located at $x = 977.0''$ and $y = -137.8''$, as marked in Extended Data Fig. 1. Solid black horizontal/vertical lines in each panel indicate the best-fit values from the GSFIT minimization. Dotted blue horizontal/vertical lines mark the median values of the MCMC probability distributions. Dashed lines in the histograms along the diagonal indicate ±1-sigma standard deviation of a given parameter. Off-diagonal panels show correlations between all possible pairs of parameters shown as 2D histograms of the probability distributions. The contours represent 39.3%, 60% and 80% of the maximum. The outer contour is selected to represent approximately the 1-sigma region of a 2D Gaussian distribution $(1 - e^{-0.5})$.

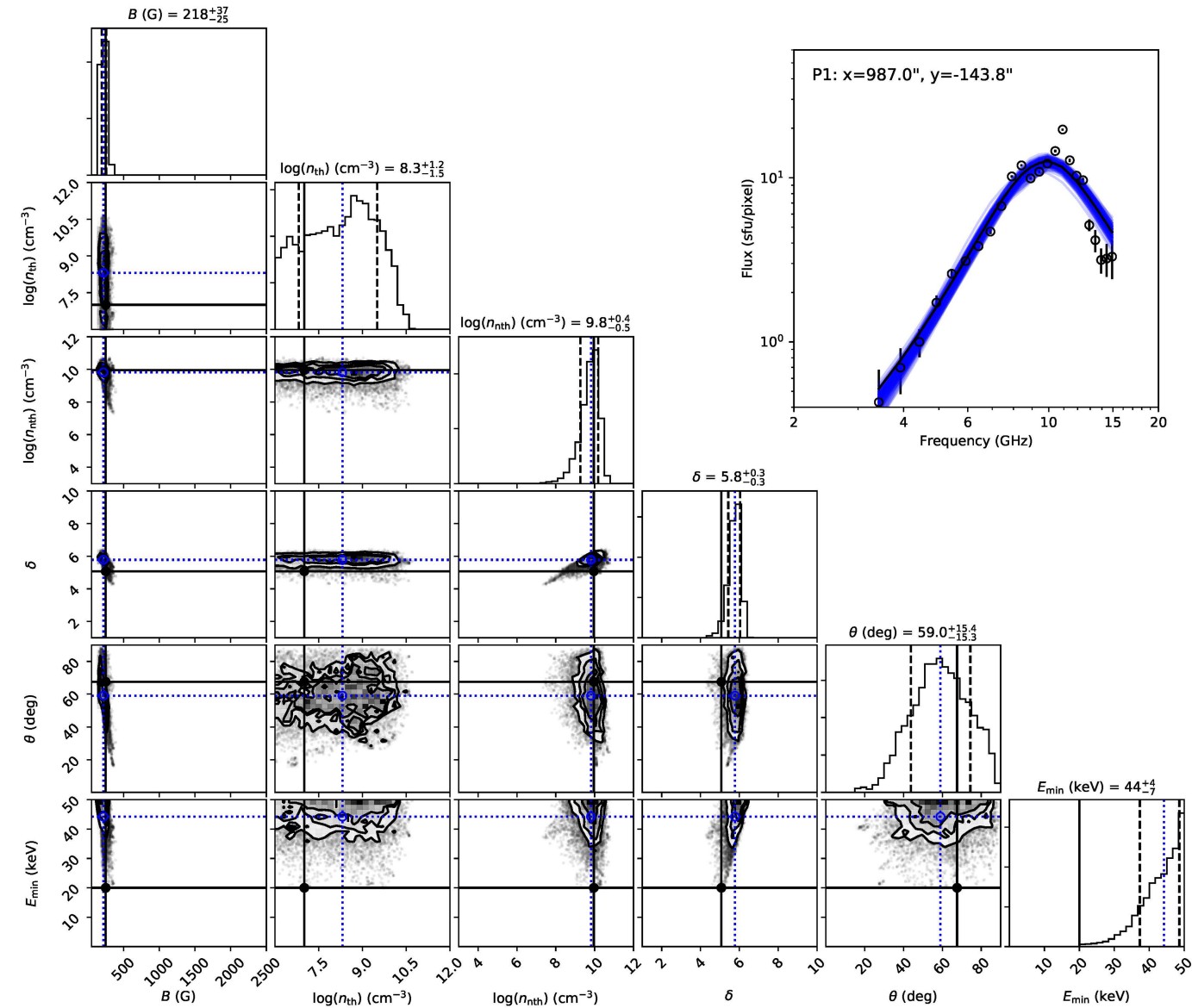

**Extended Data Fig. 3 | MCMC probability distributions of the fit parameters for an example pixel in ROI1.** The figure layout is identical to Extended Data Fig. 2 but showing the parameters for pixel P1 located at $x = 987.0''$ and $y = -143.8''$, as marked in Extended Data Fig. 1.

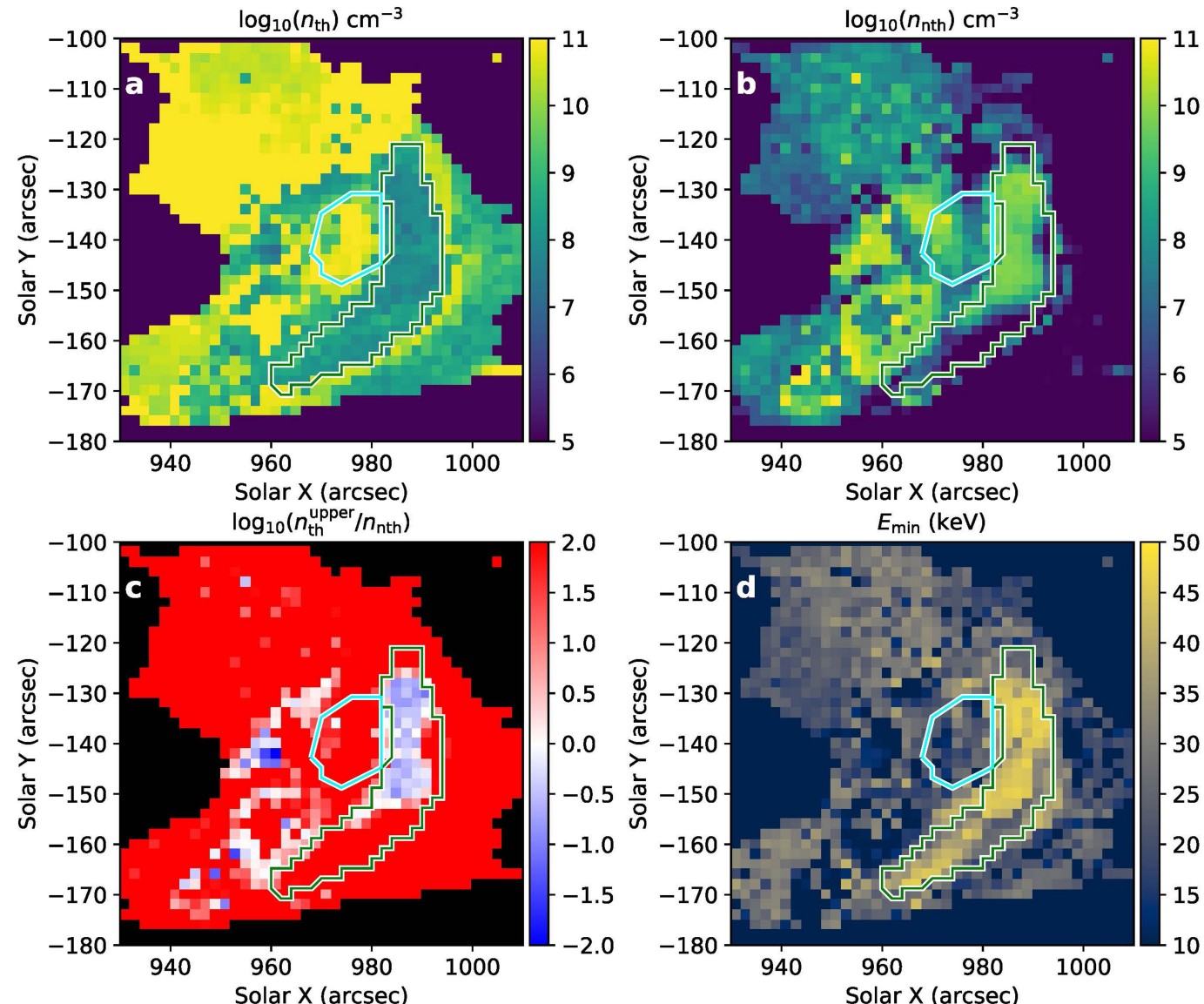

**Extended Data Fig. 4 | MCMC constrained maps of the thermal and suprathermal electron parameters. a**, Thermal electron number density $n_{th}$. For each pixel, the median value of the MCMC probability distribution is shown. **b**, Similar to **a** but for the non-thermal electron number density $n_{nth}$.

**c**, Map of the ratio of the upper limit of the thermal number density $n_{th}^{upper}$ (defined as one $\sigma$ above the median $n_{th} + \sigma_{n_{th}}$) to the non-thermal number density $n_{nth}$. **d**, Similar to **a** but for the low-energy cut-off $E_{min}$.

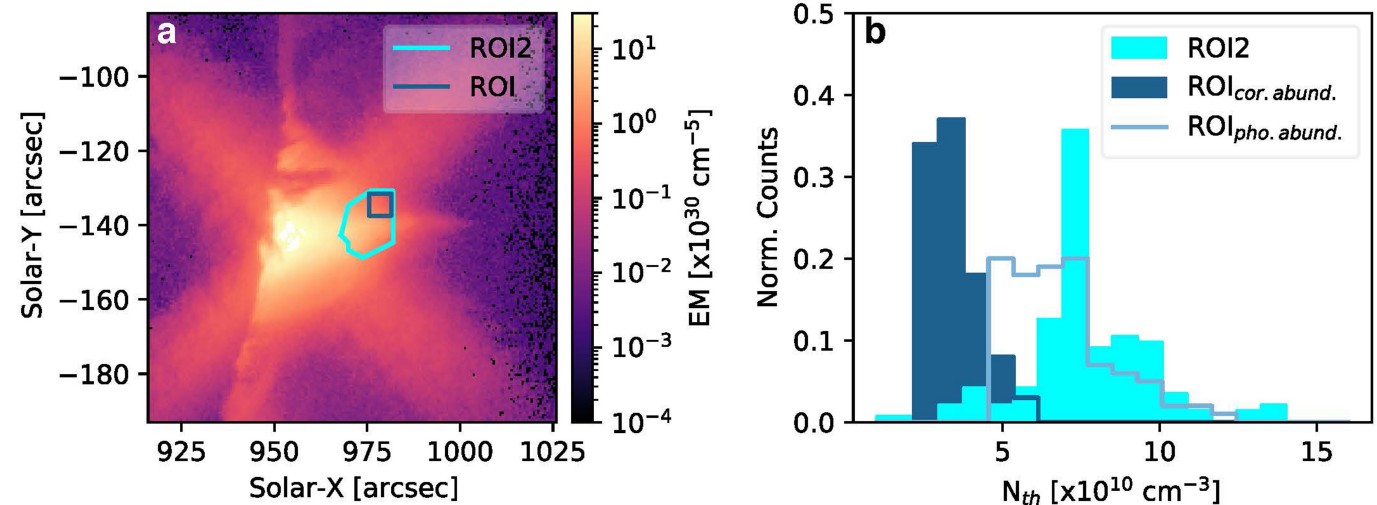

**Extended Data Fig. 5 | Comparison of thermal plasma density diagnostics. a**, EM map obtained at 15:58 UT over the temperature range 0.5–30 MK.
**b**, Histogram of the distribution of the thermal electron number density $n_{th}$ in ROI2 and the ROI obtained from microwave and EUV diagnostics, respectively.