## [Peer Review File · Nature]

Manuscript Title: Solar flare accelerates nearly all electrons in a large coronal volume

Reviewer Comments & Author Rebuttals

Reviewer Reports on the Initial Version:

Referees' comments:

Referee #1 (Remarks to the Author):

This paper analyzes data from observations of a solar flare by the Expanded Owens Valley Solar Array that were previously described in Fleishman et al., 2020 (reference 1 in this paper). Maps of electron parameters are used to investigate electron acceleration in the coronal region that showed the most prominent magnetic field decay.

Others (Krucker et al. ApJ, 2010; Krucker and Battaglia, ApJ, 2014) have previously shown that flares can accelerate essentially all electrons to nonthermal energies, so the primary novel result of this work is the observation of adjacent volumes filled with thermal and nonthermal population. The mechanism maintaining such a separation over times longer than the streaming times is unclear, but while the abstract suggests that the turbulent magnetic field is responsible, little evidence and discussion is offered in the paper.

The observation of neighboring regions which such different properties will be of interest to those in the field. However, without evidence for a physical mechanism I do not believe this paper presents sufficiently novel results for publication in Nature.

Detailed Concerns:

1. The title and abstract listed on-line are somewhat different from that in the PDF. (On-line title: "Solar Flare Features Faux Maxwell's Demon", PDF title: "A Near-Perfect Engine of Particle Acceleration in a Solar Flare".) Notably jarring is the invocation of Maxwell's demon in the title and first three sentences of the on-line version when the rest of the paper never mentions it.
2. While the abstract states that the regions of disparate populations are "maintained by the turbulent magnetic field", the text of the paper does not support this point and in fact states that (in lines 109-110) "the system contains a yet unknown but highly efficient physical process that traps the suprathermal particles". While the magnetic field seems likely to play a role, no evidence is presented to support this assertion.
3. Line 9: This first mention of magnetic energy might prove confusing for those outside the field. In particular, a brief discussion of the role of the magnetic field in triggering and powering solar flares

could let readers know why the referenced diagnostics are so important.

4. Figure 1: Are the magnetic field lines in panel a derived from a model? Somehow extracted from the data?

5. Figure 1: A curious feature in panel b is the thin line of excess thermal density just higher in altitude than ROI1. Oddly, this feature is not obvious in panel 2c (which also shows the thermal electron density), although perhaps this is a color-scale artifact. Space constraints may preclude a discussion, although it would be interesting to know whether the stripe corresponds to an expected feature in the theoretical flare picture.

6. Line 22: The text refers to ROI1 and ROI2 in the context of Figure 2, but those labels only appear in Figure 1. Would it be possible to add labels to panel 2a?

7. Lines 30-31: The text states the thermal electron number density within ROI1 is "undetectably small" in Figure 2c, but the same quantity seems to be plotted in Figure 1b. Perhaps the two panels represent data from different sources, but if so the text should be clearer.

8. Lines 98-99: While the results of $\{it\} k_{global}$ may suggest little relationship between n_{nth} and r_B when $r_B \gg 1$, as $r_B \rightarrow 1$ those same results indicate non-thermal acceleration should shut off. Are there enough data points in this regime to say that this correlation is or is not observed? (Panel 3c suggests that there are a reasonable number data points with $B_{tot}/B_{steady} < 1/\sqrt{2}$, which corresponds to equal guide and reconnecting components, but says nothing about n_{nth} .)

9. Figure S1: The caption says the figure is organized as a work chart. Is "work chart" a standard term? I'm not familiar with it (which may not mean much), but a quick search on-line did not turn up many similar examples

10. Fig. S3 and S4: It's not clear from the sparse description in the caption ("correlation between parameters") what the panels below the main diagonal are showing. In particular, what are the contour lines supposed to represent?

11. The presentation in the discussion of the MCMC validation is a bit confusing. Figure S2 is briefly mentioned (line 156), but not described in any detail until after the discussion surrounding Figures S3 and S4. Then, beginning in line 197 and extending through the rest of the section, Figure 2 is explained. Since the early mention of Figure S2 is not important to the argument -- and could be handled with a reference to another figure -- it might make sense to move the figure next to the discussion at the section's end.

12. Line 251: It is not clear why Supplemental Video S1 plots $\sqrt{n_{th}}$ and $\sqrt{n_{nth}}$ rather than just n_{th} and n_{nth} . If there is a reason, it would be helpful to note it in the caption.

13. Finally, the first sentence of the abstract ends with "[refs]" rather than a list of references. It's

also missing an "of" between "excess" and "the". (I don't comment on most grammatical and typographical details, but the exceedingly prominent placement of these two seemed to require some acknowledgment.)

Referee #2 (Remarks to the Author):

A. Summary of the key results

I found the manuscript very interesting. The manuscript reports the observation of the acceleration region and an estimation of the effectiveness of the acceleration mechanism based on EOVS radio observations of the 10 September 2021 solar flare. The authors explore the relationship between the thermal and non-thermal electron population and explain their relation based on their observations.

B. Originality and significance:

I think the work presented by the authors is novel and has major significance for the solar physics and astrophysics community.

C. Data & methodology:

The authors used a method described in Fleishman et al (2020) to calculate the density of the non-thermal population of electrons, generating density maps of the acceleration region. To validate their results, the authors used a Monte Carlo Markov Chain, finding a high degree of confidence in the calculated fitting parameters. They also studied the thermal and non-thermal electron distributions and related the results to their observations. And finally, the authors made a consistency check by comparing the microwave results with EUV thermal plasma.

D. Appropriate use of statistics and treatment of uncertainties

I think in general the statistical work is very well done. I have some objections with the results presented in Fig S1 of the Gaussian fit of the regions P1, P5 and P7. I can see that the fit is within the uncertainty, however a Gaussian is not the only solution in particular in the regions where the points diverge from the fit. This is clear in P7, where the results at high frequencies diverge, but as I mentioned above the fit is within the uncertainty.

I think the same for the results in Figure 3. In the text the authors mention that the result can be explained by a single power law, however, it can also be represented by a broken power law as the authors show in the Figure.

E. Conclusions: robustness, validity, reliability

I will urge the authors to add clear conclusions. I do understand the point of the manuscript and their main results but there is no clear conclusion.

F. Suggested improvements: experiments, data for possible revision

I think the manuscript is very good, however, there are some things that can be clarified for the general reader.

1. It is not clear the structure, does the article have sections? Seems like that, but the sudden change in figure numbering suggests that the latter sections are supplements?
2. I suggest adding some more references on particle acceleration in solar flares.
3. In line 101, could you elaborate why a larger r_B implies a smaller Δ ?
4. In lines 109-113, is your conclusion biased to the fact that the observations are 2-D projections and geometrical assumption of the model?
5. In line 142, please see my comment above on the spectra from regions P3, P5, P7 and P8. It is not clear on what stage of the flare evolution the data was taken, and since fit parameters are time functions I suggest the authors make the analysis for at least a second time interval.
6. Paragraph starting in line 201. Comparing the results in Figures 2 and S2, there is a clear difference in density range within the ROI. In Figure 2 the density values vary between $\sim 6.6\text{--}66 \times 10^9 \text{ cm}^{-3}$ and in Figure S2 the density values change between $10^5 - 10^{11}$. Am I misunderstanding something?
7. Paragraph starting in line 205. The authors mentioned that the cut-off energy they adopted is 20keV. However, they found that the median using the MCMC method is between 40 and 50 KeV. How will the results change if the authors use these values?
8. Line 220. The authors mention that they adopted a column depth of 5.8Mm for the microwave spectral fitting. Could the authors elaborate about the selection criteria for this particular column depth?
9. The argument starting in line 229 is not clear. Was the radio emission studied the addition of three frames, the median or average? Also, a region of 100 pixels in AIA is a square of 10×10 pixels, which is a 6×6 arcseconds box, which in turn is just a 3×3 pixels EOVS box. These regions seem very small, could the authors say what is beam size at the frequencies they analyzed the data?

G. References: appropriate credit to previous work?

I think the manuscript will benefit from adding some references, particularly to previous work on particle acceleration and coronal electron distributions.

H. Clarity and context:

I think the abstract needs some work, as it does not reflect clearly the contents of the manuscript. Also it seems to have some missing references.

The introduction is very good, but there is a lack of conclusions.

Author Rebuttals to Initial Comments:

Authors' response to the Referees' comments:

A: We are grateful for the thorough evaluation of our submission. Below we give our detailed responses on the comments. Additions to text other than minor stylistic or grammatical changes are given in a bold face.

Referee #1 (Remarks to the Author):

This paper analyzes data from observations of a solar flare by the Expanded Owens Valley Solar Array that were previously described in Fleishman et al., 2020 (reference 1 in this paper). Maps of electron parameters are used to investigate electron acceleration in the coronal region that showed the most prominent magnetic field decay.

R1: Others (Krucker et al. ApJ, 2010; Krucker and Battaglia, ApJ, 2014) have previously shown that flares can accelerate essentially all electrons to nonthermal energies,

A: The net result of Krucker et al. (2010) and Krucker and Battaglia (2014) is that a substantial portion of available electrons (10% or more) is being accelerated. We agree that the results of Krucker et al. (2010) and Krucker and Battaglia (2014) are fully consistent with our results, and indeed they used their observations to argue that flares can accelerate essentially all electrons to nonthermal energies, but our work goes far beyond theirs by extending our understanding of both the extent of the effect and the location and physical shape of the (sub)region in which it occurs. Hard X-ray photons require ambient density for their production; hence the location of the emission is heavily weighted towards regions of high density. In contrast the radio emission we study illuminates also low density regions to reveal the true extent of the effect. Our result is novel also in that its conclusions are clear from a single data source, microwave emission alone, with EUV data introduced only for a consistency check. Finally, we are finding that this highly efficient acceleration happens exactly where the free magnetic energy is being released – a finding also obtained from the same data set – thus, it reveals all important physical links between the magnetic energy release and associated acceleration of the charged particles.

R1: ...so the primary novel result of this work is the observation of adjacent volumes filled with thermal and nonthermal population. The mechanism maintaining such a separation over times longer than the streaming times is unclear, but while the abstract suggests that the turbulent magnetic field is responsible, little evidence and discussion is offered in the paper.

A: The separation of volumes with mainly thermal and purely nonthermal populations is interesting, but it is certainly not the primary novelty of the work. As stated above, the main novel result, obtained for the first time, is evolving spatially resolved distributions of thermal and nonthermal electrons in a solar flare. These distributions reveal a special volume filled with only (or almost only) nonthermal electrons, while being depleted of the thermal plasma. This volume is surrounded with a more usual, mainly thermal plasma mixed with a population of nonthermal electrons. We, therefore, do not claim that all electrons in the loop-top region are accelerated (in contrast to Krucker et al. ApJ, 2010; Krucker and Battaglia, ApJ, 2014 claims), but instead we report the observation of a key acceleration / energy release (sub-)region, where indeed all electrons experience a prominent acceleration, even though a lot of the thermal plasma remains elsewhere.

R1: The observation of neighboring regions which such different properties will be of interest to those in the field. However, without evidence for a physical mechanism I do not believe this paper presents sufficiently novel results for publication in Nature.

A: We agree that there was insufficient discussion of a possible physical mechanism responsible for the sustained separation of the thermal and nonthermal volumes; such a discussion is now added to the text.

Detailed Concerns:

1. The title and abstract listed on-line are somewhat different from that in the PDF. (On-line title: "Solar Flare Features Faux Maxwell's Demon", PDF title: "A Near-Perfect Engine of Particle Acceleration in a Solar Flare".) Notably jarring is the invocation of Maxwell's demon in the title and first three sentences of the on-line version when the rest of the paper never mentions it.

A: This was an unfortunate result of the Nature submission system, which did not allow changing the online version to reflect an update to the submitted manuscript before being sent out for review. This is now fixed; mentioning the Maxwell's demon has been removed to focus on the net physical findings.

2. While the abstract states that the regions of disparate populations are "maintained by the turbulent magnetic field", the text of the paper does not support this point and in fact states that (in lines 109-110) "the system contains a yet unknown but highly efficient physical process that traps the suprathermal particles". While the magnetic field seems likely to play a role, no evidence is presented to support this assertion.

A: This nuance was lost in our attempt to keep the abstract within word limits. We meant that while a general physical process responsible for the observed trapping is most likely the turbulent magnetic field and particle diffusion due to this turbulence, the details of this process remain unknown. We changed the language to make the abstract and the text consistent with each other and added more details to the text.

3. Line 9: This first mention of magnetic energy might prove confusing for those outside the field. In particular, a brief discussion of the role of the magnetic field in triggering and powering solar flares could let readers know why the referenced diagnostics are so important.

A: we agree; the missing details have been added.

4. Figure 1: Are the magnetic field lines in panel a derived from a model? Somehow extracted from the data?

A: they are drawn 'by hand' to outline the source morphology suggested by the data. This is explained in the figure caption.

5. Figure 1: A curious feature in panel b is the thin line of excess thermal density just higher in altitude than ROI1. Oddly, this feature is not obvious in panel 2c (which also shows the thermal electron density), although perhaps this is a color-scale artifact. Space constraints may preclude a discussion, although it

would be interesting to know whether the stripe corresponds to an expected feature in the theoretical flare picture.

A: the figures have been redone using the same scale and the same color tables. The presented results from the GS fit and the MCMC simulations agree with each other within the uncertainties. The figures show the corresponding mean and median values, which agree with each other where the uncertainties of the derived parameters are small. In the outward region the uncertainties get larger, which is the ground of the noted mismatch between the figures. The mentioned thin line is there (in panel 2c), but masked by a more extended yellowish area in the most right portion of the figure, where, as has been said, uncertainties of the derived parameters are large. The clarification is added to the text.

6. Line 22: The text refers to ROI1 and ROI2 in the context of Figure 2, but those labels only appear in Figure 1. Would it be possible to add labels to panel 2a?

A: thanks; fixed.

7. Lines 30-31: The text states the thermal electron number density within ROI1 is "undetectably small" in Figure 2c, but the same quantity seems to be plotted in Figure 1b. Perhaps the two panels represent data from different sources, but if so the text should be clearer.

A: we believe this issue is removed by replotting the figures mentioned above by using the same scaling and color table and by adding a clarification in Figure 2 caption.

8. Lines 98-99: While the results of $\{it\} \text{kglobal}$ may suggest little relationship between n_{nth} and r_B when $r_B \gg 1$, as $r_B \rightarrow 1$ those same results indicate non-thermal acceleration should shut off. Are there enough data points in this regime to say that this correlation is or is not observed? (Panel 3c suggests that there are a reasonable number data points with $B_{tot}/B_{steady} < 1/\sqrt{2}$, which corresponds to equal guide and reconnecting components, but says nothing about n_{nth} .)

A: we investigated this question. We only find some correlation if we include the data points with both large and small number density of the nonthermal population, but this is beyond the scope of our study. If we restrict the r_B at small numbers around 1-1.5, there is insufficient range to derive any meaningful regression. There are many data points with relatively small r_B , where nonthermal population dominates. This likely indicates that even when acceleration is over, the accelerated electrons remain trapped in the region.

9. Figure S1: The caption says the figure is organized as a work chart. Is "work chart" a standard term? I'm not familiar with it (which may not mean much), but a quick search on-line did not turn up many similar examples

A: this confusing sentence has been removed.

10. Fig. S3 and S4: It's not clear from the spare description in the caption ("correlation between parameters") what the panels below the main diagonal are showing. In particular, what are the contour lines supposed to represent?

A: the figure caption has been expanded to give all relevant information.

11. The presentation in the discussion of the MCMC validation is a bit confusing. Figure S2 is briefly mentioned (line 156), but not described in any detail until after the discussion surrounding Figures S3 and S4. Then, beginning in line 197 and extending through the rest of the section, Figure 2 is explained. Since the early mention of Figure S2 is not important to the argument -- and could be handled with a reference to another figure -- it might make sense to move the figure next to the discussion at the section's end.

A: the section has been revised accordingly.

12. Line 251: It is not clear why Supplemental Video S1 plots $\sqrt{n_{th}}$ and $\sqrt{n_{nth}}$ rather than just n_{th} and n_{nth} . If there is a reason, it would be helpful to note it in the caption.

A: we recreated the video using log scale, which is consistent with figures in the text.

13. Finally, the first sentence of the abstract ends with "[refs]" rather than a list of references. It's also missing an "of" between "excess" and "the". (I don't comment on most grammatical and typographical details, but the exceedingly prominent placement of these two seemed to require some acknowledgment.)

A: the missing references have been added.

Referee #2 (Remarks to the Author):

A. Summary of the key results

I found the manuscript very interesting. The manuscript reports the observation of the acceleration region and an estimation of the effectiveness of the acceleration mechanism based on EOVS radio observations of the 10 September 2021 solar flare. The authors explore the relationship between the thermal and non-thermal electron population and explain their relation based on their observations.

B. Originality and significance:

I think the work presented by the authors is novel and has major significance for the solar physics and astrophysics community.

C. Data & methodology:

The authors used a method described in Fleishman et al (2020) to calculate the density of the non-thermal population of electrons, generating density maps of the acceleration region. To validate their results, the authors used a Monte Carlo Markov Chain, finding a high degree of confidence in the calculated fitting parameters. They also studied the thermal and non-thermal electron distributions and related the results to their observations. And finally, the authors made a consistency check by comparing the microwave results with EUV thermal plasma.

D. Appropriate use of statistics and treatment of uncertainties

I think in general the statistical work is very well done. I have some objections with the results presented in Fig S1 of the Gaussian fit of the regions P1, P5 and P7. I can see that the fit is within the uncertainty, however a Gaussian is not the only solution in particular in the regions where the points diverge from the fit. This is clear in P7, where the results at high frequencies diverge, but as I mentioned above the fit is within the uncertainty.

A: We show the spectra (from the corners) for a reference only. Not only the data uncertainties but also the uncertainties of the physical parameters derived from the fit are large there. To be clear, the fits to the data are not Gaussian fits, but rather fits of theoretical gyrosynchrotron spectra from a homogeneous source. Such theoretical spectra are constrained by the parameters, within the uncertainties of the individual data points, and the displayed fits are the ones with minimum chi-square. Because of the large uncertainties in the parameters, however, they are not used in the quantitative analysis or to draw any physical conclusion. As we have said, we only consider good spectra and fits from ROI I and ROI II.

I think the same for the results in Figure 3. In the text the authors mention that the result can be explained by a single power law, however, it can also be represented by a broken power law as the authors show in the Figure.

A: we only show the single power-law. We agree that there can be other fits consistent with the data in this figure. The linear cross-correlation is used to show the presence of the revealed trend.

E. Conclusions: robustness, validity, reliability

I will urge the authors to add clear conclusions. I do understand the point of the manuscript and their main results but there is no clear conclusion.

A: thanks! The conclusions have been added.

F. Suggested improvements: experiments, data for possible revision

I think the manuscript is very good, however, there are some things that can be clarified for the general reader.

1. It is not clear the structure, does the article have sections? Seems like that, but the sudden change in figure numbering suggests that the latter sections are supplements?

A: yes, the Method (online supplement) Section starts at line 114 of the original submission.

2. I suggest adding some more references on particle acceleration in solar flares.

A: thanks; added.

3. In line 101, could you elaborate why a larger r_B implies a smaller δ ?

A: it is, in fact, explained there: **“A simplistic interpretation of this relationship is that having more free magnetic energy (larger r_B) permits acceleration to higher energies, thus, producing a flatter distribution of the accelerated electrons over energy.”**

4. In lines 109-113, is your conclusion biased to the fact that the observations are 2-D projections and geometrical assumption of the model?

A: we rephrased this sentence to make it clearer.

5. In line 142, please see my comment above on the spectra from regions P3, P5, P7 and P8. It is not clear on what stage of the flare evolution the data was taken, and since fit parameters are time functions I suggest the authors make the analysis for at least a second time interval.

A: we made the analysis for all available time frames, but show only a subset of this analysis in the representative time frame. We clarified this in the text.

6. Paragraph starting in line 201. Comparing the results in Figures 2 and S2, there is a clear difference in density range within the ROI. In Figure 2 the density values vary between $\sim 6.6\text{--}66 \times 10^9 \text{ cm}^{-3}$ and in Figure S2 the density values change between $10^5 - 10^{11}$. Am I misunderstanding something?

A: this issue has been fixed by new versions of the figures and their updated captions.

7. Paragraph starting in line 205. The authors mentioned that the cut-off energy they adopted is 20keV. However, they found that the median using the MCMC method is between 40 and 50 KeV. How will the results change if the authors use these values?

A: that high values of the low-energy cut-off, 40-50 keV, are only detected in ROI I, where both GSFIT and MCMC approaches reveal that all available electrons are accelerated to nonthermal energies. MCMC approach reveals acceleration to higher energies, which only strengthens the finding. We added a discussion of this important issue in the Methods section, in particular, “... n_{nth} does not correlate with E_{min} ; thus, the conclusion of the high nonthermal number density is robust...”

8. Line 220. The authors mention that they adopted a column depth of 5.8Mm for the microwave spectral fitting. Could the authors elaborate about the selection criteria for this particular column depth?

A: this corresponds to 8” on disk – this is a scale of features (loops) seen in the flare images. We have included a statement motivating our choice in the updated figure caption.

9. The argument starting in line 229 is not clear. Was the radio emission studied the addition of three frames, the median or average? Also, a region of 100 pixels in AIA is a square of 10x10 pixels, which is a 6x6 arcseconds box, which in turn is just a 3x3 pixels EOVSAs box. These regions seem very small, could the authors say what is beam size at the frequencies they analyzed the data?

A: we made clearer that we added data from three consecutive time frames separated by 4 seconds to compare with 12-second AIA cadence. We consider two different boxes for radio and AIA – just for the reason indicated by the reviewer – AIA 6x6 arcseconds box is just a 3x3 pixels EOVSAs box, which is too small. Thus, we use a larger box for the EOVSAs data to make our statistical comparison (not a pixel-by-pixel one). We added information about the beam size.

G. References: appropriate credit to previous work?

I think the manuscript will benefit from adding some references, particularly to previous work on particle acceleration and coronal electron distributions.

A: thanks; done!

H. Clarity and context:

I think the abstract needs some work, as it does not reflect clearly the contents of the manuscript. Also it seems to have some missing references.

A: The abstract has been modified, and missing references have been added.

The introduction is very good, but there is a lack of conclusions.

A: conclusions have been added.

Reviewer Reports on the First Revision:

Referees' comments:

Referee #1 (Remarks to the Author):

The authors have significantly improved the manuscript in response to the comments from the reviewers.

The description of the statistical procedures has been improved and seems mostly complete. It would be helpful if the caption for Figure S1 explicitly stated that the error bars are 1-sigma values, although that seems to be the implication.

I (and the other referee) raised concerns about the lack of conclusions. The additions to the revised paper address this issue and, in my opinion, significantly strengthen the paper.

My primary concern with the previous version was novelty, namely whether the work's conclusions significantly differed from previous X-ray observations of nonthermal particle acceleration in solar flares. The authors make a strong case for the originality of this work in their reply and, after reviewing the papers in question (Krucker et al. (2010); Krucker and Battaglia (2014)), I was mostly convinced. However much of this justification -- which would help place this work in context for many readers -- does not appear in the papers, perhaps due to space limitations. Instead, the text suggests the previous work has been "put into question" with a single reference to a (presumably) non-peer-reviewed AAS abstract. Such a statement does the paper no favors and almost certainly does not represent the sentiments of the broader community. I encourage the authors to instead include as much of the justification that appeared in the rebuttal as possible. Space is at a premium, of course, but establishing why the results matter is at least as important as the results themselves.

Scientifically I think this work represents one of the top few solar physics papers of the last year and could reasonably be published in Nature. However, I think some relatively minor improvements to the justification and context of the work would enhance its ultimate impact.

Referee #2 (Remarks to the Author):

I found the manuscript very interesting. The manuscript reports the observation of the acceleration region and an estimation of the effectiveness of the acceleration mechanism based on EOVS radio observations of the 10 September 2021 solar flare. The authors explore the relationship between the thermal and non-thermal electron population and explain their relation based on their observations.

I think the work presented by the authors is novel and has major significance for the solar physics and astrophysics community.

The authors used a method described in Fleishman et al (2020) to calculate the density of the non-thermal population of electrons, generating density maps of the acceleration region. To validate

their results, the authors used a Monte Carlo Markov Chain, finding a high degree of confidence in the calculated fitting parameters. They also studied the thermal and non-thermal electron distributions and related the results to their observations. And finally, the authors made a consistency check by comparing the microwave results with EUV thermal plasma.

The conclusion reflects the observation of spatially resolved regions of free magnetic energy release during a solar flare. The authors conclude that this release of energy is capable of accelerating all available electrons in regions studied to high non-thermal energies. The authors suggest that a highly turbulent state of the ambient magnetic field enhances spatial diffusion of the charged particles leading to a more effective trapping and acceleration of the electrons.

Author Rebuttals to First Revision:

Referees' comments:

Referee #1 (Remarks to the Author):

The authors have significantly improved the manuscript in response to the comments from the reviewers.

The description of the statistical procedures has been improved and seems mostly complete. It would be helpful if the caption for Figure S1 explicitly stated that the error bars are 1-sigma values, although that seems to be the implication.

A: added.

I (and the other referee) raised concerns about the lack of conclusions. The additions to the revised paper address this issue and, in my opinion, significantly strengthen the paper.

My primary concern with the previous version was novelty, namely whether the work's conclusions significantly differed from previous X-ray observations of nonthermal particle acceleration in solar flares. The authors make a strong case for the originality of this work in their reply and, after reviewing the papers in question (Krucker et al. (2010); Krucker and Battaglia (2014)), I was mostly convinced. However much of this justification -- which would help place this work in context for many readers -- does not appear in the papers, perhaps due to space limitations. Instead, the text suggests the previous work has been "put into question" with a single reference to a (presumably) non-peer-reviewed AAS abstract. Such a statement does the paper no favors and almost certainly does not represent the sentiments of the broader community. I encourage the authors to instead include as much of the justification that appeared in the rebuttal as possible. Space is at a premium, of course, but establishing why the results matter is at least as important as the results themselves.

A: this comment has fully been taken into account; the Summary has been rewritten accordingly.

Scientifically I think this work represents one of the top few solar physics papers of the last year and could reasonably be published in Nature. However, I think some relatively minor improvements to the justification and context of the work would enhance its ultimate impact.

Referee #2 (Remarks to the Author):

I found the manuscript very interesting. The manuscript reports the observation of the acceleration region and an estimation of the effectiveness of the acceleration mechanism based on EOVS radio observations of the 10 September 2021 solar flare. The authors explore the relationship between the thermal and non-thermal electron population and explain their relation based on their observations.

I think the work presented by the authors is novel and has major significance for the solar physics and astrophysics community.

The authors used a method described in Fleishman et al (2020) to calculate the density of the non-thermal population of electrons, generating density maps of the acceleration region. To validate their results, the authors used a Monte Carlo Markov Chain, finding a high degree of confidence in the calculated fitting parameters. They also studied the thermal and non-thermal electron distributions and related the results to their observations. And finally, the authors made a consistency check by comparing the microwave results with EUV thermal plasma.

The conclusion reflects the observation of spatially resolved regions of free magnetic energy release during a solar flare. The authors conclude that this release of energy is capable of accelerating all available electrons in regions studied to high non-thermal energies. The authors suggest that a highly turbulent state of the ambient magnetic field enhances spatial diffusion of the charged particles leading to a more effective trapping and acceleration of the electrons.

Gregory Fleishman

Reviewer Reports on the Second Revision:

Referees' comments:

Referee #1 (Remarks to the Author):

The re-revised version of the manuscript has been significantly improved. In my view it represents one of the top few solar physics papers of the last year and should be published in Nature.

The primary concern of my last review was the text's first few paragraphs that should have, ideally, laid out the relevant issues for a general reader. I'm happy to report that the authors have made a concerted, and to my mind successful, effort to improve this section. The importance of the work and the advances over previous research are clearly communicated in a way that should pique the interest of anyone flipping (scrolling?) through the pages. The remainder of the text has also been tightened up, yet still addresses the issues raised in the previous rounds of review. The description of the statistical procedures seems to be complete. Finally, the addition of a new video to the Supplementary Material and a few other tweaks also represent improvements.

In short, I recommend the paper for publication.